# A multi-omic meta-analysis reveals novel mechanisms of insecticide resistance in malaria vectors
Sanjay C. Nagi [1] ✉ & Victoria A. Ingham [2,3] ✉

Malaria control faces challenges from widespread insecticide resistance in major *Anopheles* species. This study, employing a cross-species approach, integrates RNA-Sequencing, whole-genome sequencing, and microarray data to elucidate drivers of insecticide resistance in *Anopheles gambiae* complex and *An. funestus*. Here we show an inverse relationship between genetic diversity and gene expression, with highly expressed genes experiencing stronger purifying selection. Gene expression clusters physically in the genome, revealing potential coordinated regulation, and we find that highly over-expressed genes are associated with selective sweep loci. We identify known and novel candidate insecticide resistance genes, enriched for metabolic, cuticular, and behavioural functioning. We also present AnoExpress, a Python package, and an online interface for user-friendly exploration of resistance candidate expression. Despite millions of years of speciation, convergent gene expression responses to insecticidal selection pressures are observed across *Anopheles* species, providing crucial insights for malaria vector control.

Malaria remains a leading cause of worldwide morbidity and mortality, resulting in an estimated 619,000 deaths in 2021[1]. Insecticide-based vector control interventions are the most successful tools for reducing malaria incidence, responsible for over 80% of the malaria cases averted between 2000 and 2015[2]; however, the gains made in malaria control have plateaued in the last eight years and cases are beginning to rise[1]. The reversal of gains is partially due to widespread insecticide resistance within populations of the anopheline mosquito vector[3], reducing the efficacy of insecticide-treated bed neds (ITNs) and indoor residual spraying (IRS), the two primary malaria intervention tools[1,4].

Across sub-Saharan Africa, four mosquito species are responsible for the majority of malaria transmission, the *An. gambiae* species complex (*An. gambiae, An. arabiensis,* and *An. coluzzii*) and *An. funestus*[5–7]. Understanding the similarities and differences in resistance mechanisms between the different dominant vector species is critical to inform malaria mitigation strategies, which are mostly employed indiscriminately despite differences in vector composition between and within areas of sub-Saharan Africa.

Sharing of resistance mechanisms between insects is commonly observed; mutations at the insecticide target site genes *Rdl*, *Kdr* and *Ace-1* occur at equivalent codons throughout arthropods[8,9], and insecticide detoxification is often driven by cytochrome P450s from the insect CYP6 family[10]; this may occur through parallel evolution, or in closely related

insects, introgression, where genetic material is passed between species via hybridisation. Although adaptive introgression is seen at resistance loci within the *An. gambiae* species complex[11,12], there is no evidence of hybridisation or introgression with *An. funestus*, which diverged approximately 80 million years ago[13,14].

Insecticide resistance in *Anopheles* mosquitoes commonly involves large gene families, such as cytochrome P450 monooxygenases (P450s), glutathione S-transferases (GSTs), carboxylesterases (COEs), and UDP-glycosyltransferases (UDPs)[15–17]. Upregulation of key P450s has been reported in insecticide resistant populations of each *Anopheles* species, for example *CYP6Z1, CYP9K1, CYP6AA1* and *CYP6P9a* in *An. funestus*[18–20]; *CYP9K1, CYP6AA1, CYP6P3* and *CYP6M2* in *An. gambiae* and *An. coluzzii*[21–23]; and *CYP6P4* in *An. arabiensis*[24]. Similarly, cuticular resistance[25], which involves thickening and modification of the composition of the cuticle, slowing the rate of absorption of insecticides has been reported in both *An. gambiae s.l*[26] and *An. funestus*[27]. Work comparing distinct *Anopheles* species found that gene numbers within these families were similar across species but lineage-specific losses and gains are regularly observed[14]. Despite these similarities, differences between the *An. gambiae* species complex and *An. funestus* are also observed. A prominent example is single nucleotide polymorphisms (SNPs) in the target site of insecticide, which reduce the efficacy of insecticides through changes to binding affinity.

[1]Vector Biology Department, Liverpool School of Tropical Medicine, Liverpool, UK. [2]Centre for Infectious Diseases, University Hospital Heidelberg, Medical Faculty, Heidelberg University, Heidelberg, Germany. [3]German Center for Infection Research (DZIF), Partner Site Heidelberg, Heidelberg, Germany. ✉e-mail: Sanjay.nagi@lstmed.ac.uk; Victoria.ingham@uni-heidelberg.de

The two best studied are *Kdr* and *Ace-1*[28,29], both of which are found in the *An. gambiae* species complex, but which are seemingly absent in *An. funestus*. Novel mechanisms of resistance have also recently been discovered, including the direct binding of pyrethroids by chemosensory proteins (CSPs)[30], as well as evidence for a role of both hexamerins, α-crystallins[31]. The recent discovery of such mechanisms highlights the complexity of resistance and the need for further study into molecular mechanisms of resistance in major malaria vectors.

A plethora of 'omics data has now been generated for malaria vectors. Over the past two decades, transcriptomic studies have driven discoveries into mechanisms of insecticide resistance, first in the form of microarrays, and subsequently RNA-Sequencing. In addition, since publishing the first phase of the Anopheles 1000 genomes project in 2017[32], the MalariaGEN Vector Observatory has generated and made public thousands of high-quality whole-genome sequences of major malaria vectors from throughout sub-Saharan Africa. Although individual -omics experiments have had great success in identifying highly over-expressed transcripts involved in resistance, these studies result in thousands of differentially expressed genes resulting from noise from the susceptible comparator. Previous work has successfully used microarray data in exploratory analyses to identify patterns of gene expression across the *An. gambiae* species complex, identifying meta-signatures across geographically and temporally disparate data, highlighting the importance of meta-analyses[30,31,33]. In this study we characterise gene-expression profiles from published transcriptomic studies across two malaria vector complexes and establish relationships with whole-genome sequence data. To achieve this, we explored expression in resistance candidate genes and protein-coding families across 28 individual experiments of *An. gambiae s.l* and *An. funestus*[34–40]. These data have been integrated both with the previous microarray meta-analysis[31] and with whole-genome sequence data from the Anopheles 1000 genome project[32]. The meta-analysis performed here has been made available as a resource to the community as a user-friendly python package, AnoExpress, combined with convenient interactive notebooks intended to be run in Google Colaboratory.

## Results
### AnoExpress
We performed a differential expression meta-analysis on read count data from 28 RNA-sequencing datasets representing insecticide-resistant *Anopheles* populations from 11 countries in sub-Saharan Africa[34–40] (Fig. 1, Supplementary Table 1). Twenty-eight resistant populations were included

in this study from *An. coluzzii* (n = 15), *An. gambiae* (n = 3) *An. arabiensis* (n = 5) and *An. funestus* (n = 5). Read-count data from different species were derived from aligning to each respective reference genome assembly. For differential expression analysis, each resistant population was compared to an insecticide-susceptible strain of the same species from the same study with DESeq2. As read-counting methods differed between published studies, we first performed an experiment with RNA-Sequencing data from Nagi et al., (2023)[41] to ensure that fold-change data derived from these methods would be comparable. We compared three read-counting methods, HISAT2-featurecounts, HISAT2-htseqcount, and Kallisto, and found high pearsons R correlations of log2 fold-change values between methods (all > 0.922) (Supplementary Text 1, Supplementary Table 2). Next, to ensure that different methodologies such as fragment vs read count did not significantly influence the results, we performed a correlation test (Supplementary Text 1, Supplementary Fig. 1) and found extremely high correlations of both differentially expressed genes (0.9667) and normalised counts (0.9997). To enable cross-species comparison of putative resistance genes, we located orthologs between each different genome assembly and the AgamP4 PEST reference genome, using the OrthoMCL algorithm in VectorBase. As the inclusion of each successive assembly reduces the numbers of genes that are present, we provide four differential expression analyses; the full dataset, termed '*gamb_colu_arab_fun*' (8599 genes), and secondary analyses – '*gamb_colu_arab*' (8651 genes), '*gamb_colu*' (11,288 genes), '*fun*' (14,176 genes). Analyses herein were performed on the full *gamb_colu_arab_fun* dataset unless specified.

These data are presented within a python package, AnoExpress, made for the community, with example notebooks provided to run in Google Colab. Users can load, explore, and visualise the gene expression meta-analysis data, including reproducing most analyses presented herein. AnoExpress can also directly load gene expression data from an earlier meta-analysis of microarray data[31]. AnoExpress is located here: github.com/sanjaynagi/AnoExpress. Documentation and a video user-guide are provided to improve ease of access.

### Dataset structure
To investigate overall structure in the dataset, we performed principal components analysis on both the count and fold-change data. PCA on the log-transformed count data revealed five distinct clusters present. Four clusters contain samples only from *An. funestus*, *An. coluzzii*, or *An. arabiensis*, with a fifth cluster containing a mixture of *An. gambiae*, *An. arabiensis* and *An. coluzzii* (Fig. 2A). The four species did not resolve into four

**Fig. 1 | Overview of collection location for resistant populations in each RNA-Sequencing study.** Different colours show different species. Sample sites with multiple colours indicate multiple species comparisons at that location. Information on the susceptible comparator species is available in Supplementary Table 1. The map is open source and is produced with geopandas, contextily, cartopy and inkscape.

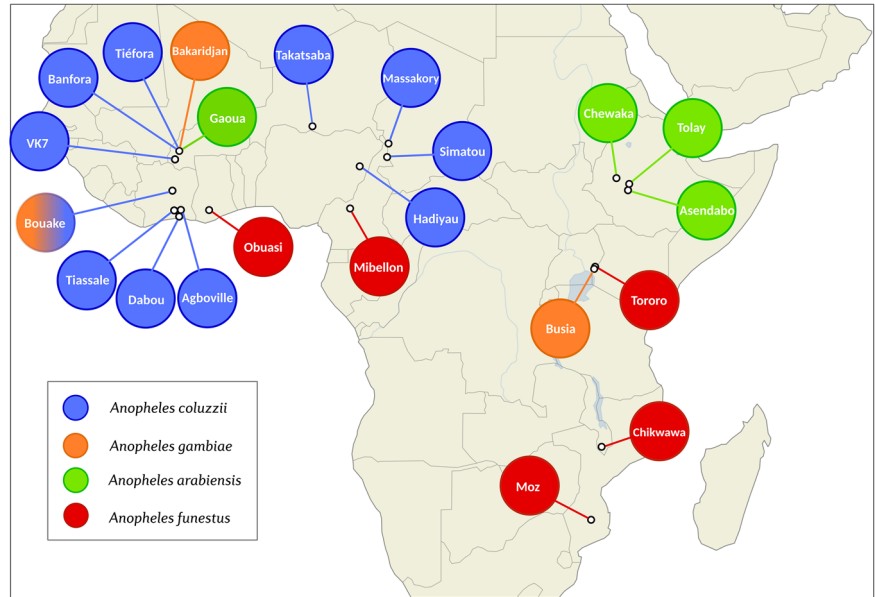

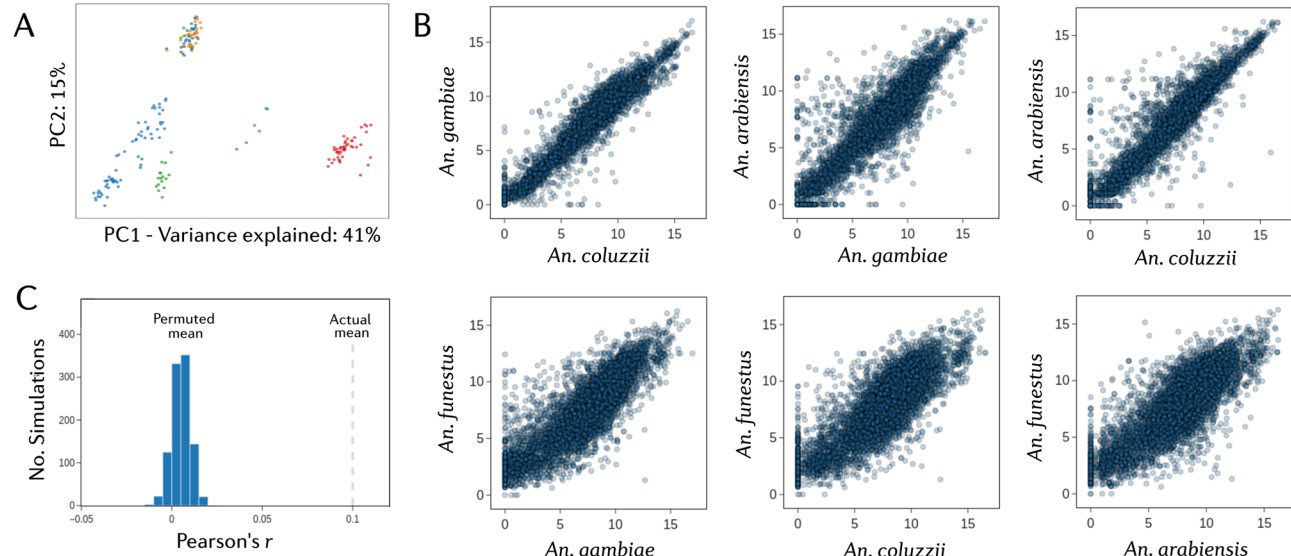

**Fig. 2 | Structure of the transcriptomic meta-analysis dataset. A** A principal components analysis of normalised log2 read count data from the AnoExpress differential expression meta-analysis. *An. gambiae = orange, An. coluzzii = blue, and An. arabiensis = green.* **B** Per-gene correlations of normalised log2 read count data between species. **C** A histogram of mean pearsons R values of log2 fold change between neighbouring genes, from 1000 random permutations of genome position, with the true value plotted as a dashed vertical line.

distinct clusters, suggesting that batch effects from different studies could be present. We therefore only performed within-study differential expression analysis. To compare RNA-Sequencing data with the earlier microarray studies, we performed a PCA on the fold-change data from all available experiments (Fig. 2A, Supplementary Figs. 2A-C). There was no evidence of clustering between the two technologies, suggesting that the results between the two methodologies are comparable and can be used in combined analyses.

Correlations between pairs of orthologous genes count data from different species were high, with lower correlation between *An. funestus* and the other three species, which is to be expected given their considerable divergence time (Fig. 2B).

## Gene expression is physically clustered in the genome

As changes in gene expression are likely to be related to cis-acting factors and the local genetic context[42], genes in close physical proximity are more likely to show similar patterns of expression. To test this within our dataset, we calculated and averaged the Pearson's correlation on log2 fold-changes between each neighbouring gene in the dataset, resulting in a mean Pearson's R of 0.10 for neighbouring genes. To determine whether this value was higher than expected based on chance, we randomly permuted the positions of all genes in the genome 1000 times and re-calculated the mean Pearson's R for now neighbouring genes. Figure 2C shows the histogram of mean Pearson's correlations, demonstrating a clear effect of genome proximity on gene expression.

To further explore physical clustering of resistance-related gene expression, we visualised fold-change values across the whole genome. In Fig. 3, we use AnoExpress to plot gene expression for all genes across all comparisons in the *gamb_colu_arab_fun* analysis and calculate the moving average log2 fold change across the genome in sliding windows. We identify gene clusters known to be associated with insecticide resistance, such as the *CYP6P, CYP9K, CYP6M/Z* and *GSTD* gene clusters. This analysis also highlighted numerous clusters of cuticle proteins which show elevated expression.

## Genetic diversity and purifying selection scale with levels of gene expression

We postulated that genes showing higher levels of absolute expression are more likely to be functional in multiple tissues or pathways, and so should be highly conserved, experiencing the highest levels of purifying selection. We integrated whole-genome sequence data from the Anopheles 1000 genomes project[32], to explore the relationship between gene expression and genetic diversity, for each species in which genomic data was publicly available. To compare between species, we selected representative cohorts of each species from across the species' range; *An. gambiae* from Burkina Faso, Democratic Republic of Congo, and Uganda), *An. coluzzii* from Burkina Faso, Cote D'Ivoire, and Kenya), and *An. arabiensis* from Burkina Faso, Tanzania, and Uganda), collected between 2012 and 2019. These cohorts show no signatures of inbreeding or major demographic events. As a measure of absolute expression, we used normalised count data. We then counted the number of segregating sites of synonymous and non-synonymous mutations, as well as calculating nucleotide diversity, π, in each gene, for each species.

Figure 4 shows the ratio of pN/pS and nucleotide diversity, for various levels of expression binned into 5% percentiles for the *gamb_colu_arab* analysis. We show that average nucleotide diversity is reduced for the most highly expressed genes and is lower overall in *An. arabiensis* compared to *An. coluzzii* and *An. gambiae*, which fits with expectations on genetic diversity from the literature[43].

We show that for each species, the pN/pS ratio is lower for the most highly expressed genes compared to lowly expressed genes. This holds more strongly in *An. gambiae* and *An. coluzzii* than *An. arabiensis*, which likely relates to the efficiency of purifying selection depending on population size —the effects of purifying selection are expected to be stronger in larger populations.

## Highly overexpressed genes are associated with selective sweeps in wild-caught mosquitoes

Given that insecticide resistance is often associated with increased expression of metabolic genes and that beneficial resistance mutations can cause selective sweeps which spread through a population, we hypothesised that highly expressed genes are more likely to be found in proximity to genomic regions under selection in wild-caught mosquitoes.

To determine this, we investigated genome-wide signals of recent selection using the H12 statistic[44] on data from phase 3 of the Anopheles 1000 genomes project (Nagi et al., in prep). Every signal was then mined to determine which genes in the AgamP4 assembly, if any, lie at the location of the peak of a selection signal. Further information on

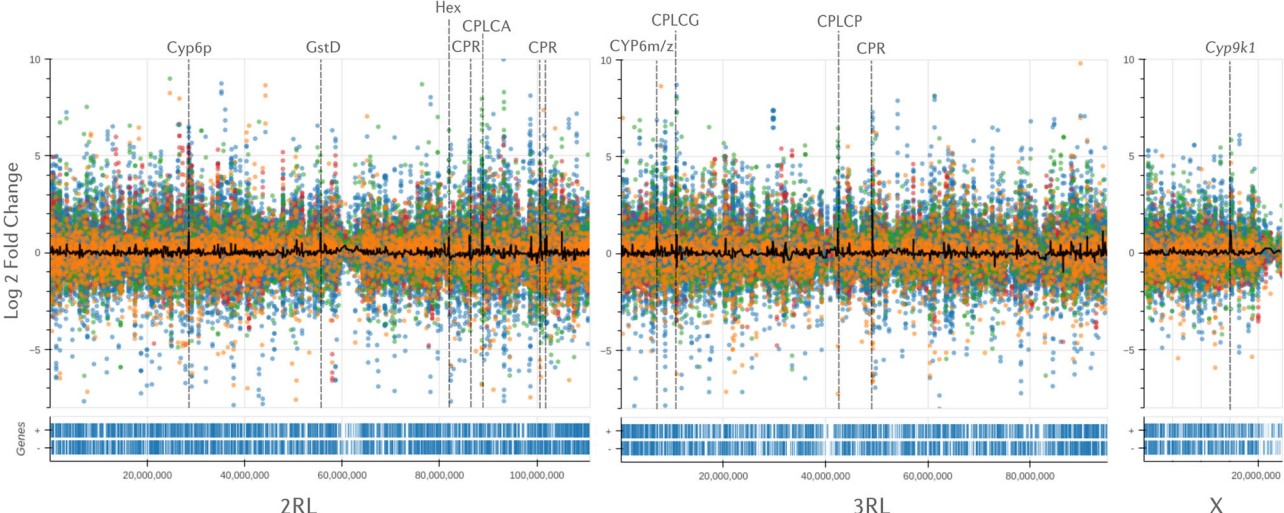

**Fig. 3 | Genome-wide expression scans.** Expression for all genes in the *gamb_colu_arab_fun* analysis is plotted against the *An. gambiae* PEST reference genome. Different colours represent different species – blue = *An. coluzzii*, orange = *An. gambiae*, green = *An. arabiensis* and red = *An. funestus*. A moving average of gene expression is plotted as a black line, calculated in sliding windows of 10 genes, with a step of 2 genes. The Y-axis is truncated between -8 and +10 log2 Fold-change to ease interpretation. Signals in close proximity to known, or putative, IR loci are labelled and highlighted with dashed grey lines. Gene density is displayed below the x-axis. In AnoExpress, plots are interactive, aiding interpretation.

**Fig. 4 | Gene Expression shapes levels of purifying selection and nucleotide diversity across *Anopheles* species. A** The ratio of non-synonymous to synonymous segregating sites (pN/pS) in *An. coluzzii* = blue, *An. gambiae* = orange, and *An. arabiensis* = green, at different levels of gene expression. **B**) Nucleotide diversity, π, calculated across the entire length of a gene. Levels of gene expression are based on median log2 counts from the *gamb_colu_arab* analysis.

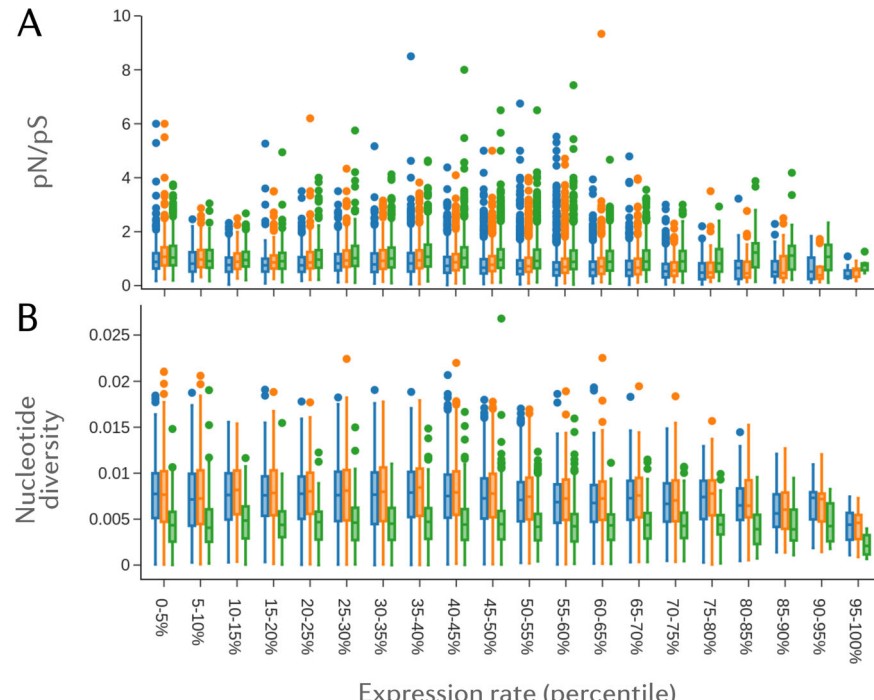

genome-wide selection scans and signal calling is found in Supplementary Text 2. Candidate genes were then defined as those genes showing a median fold-change of greater than two in the *An. gambiae, An. coluzzii* and *An. arabiensis* dataset *(gamb_colu_arab)*, as these three species are represented in the Ag1000G phase 3 data[32]. We additionally filtered out lowly expressed genes (median counts < 5). After filtering, 106 candidate genes remained, of which 34 were located at the site of a signal of selection (Supplementary Table 3). Figure 5 displays the location of these genes in the *An. gambiae* PEST reference genome, and recovers known insecticide resistance candidates such as *CYP6P3, CYP6M2, GSTE2* and *CYP9K1*, as well as numerous other genes putatively involved in resistance.

To determine whether this was a significant enrichment of genes compared to random chance, we carried out a permutation analysis. In 10,000 permutations, only two equalled or exceeded the value of 34 (p-value = 0.0002), suggesting that regions of the genome under selection are indeed enriched for the most highly overexpressed genes. Separately, we utilised the entire dataset of genes and found a positive association between a genes' median fold-change and whether it was found at the site of an H12 selective sweep signal ($p_{GLM}$ = 0.000).

## Signatures of resistance-associated gene expression
To explore highly over-expressed genes across all four species, we ranked genes by mean and median log$_2$ fold change across all experiments

**Fig. 5 | Insecticide resistance-associated selection and expression in *An. gambiae*.** The location of genes in the AgamP4 PEST reference genome, which are both found in regions of selective sweeps in whole-genome sequence data, and which are also resistance candidates based on average fold-changes in the RNA-Sequencing meta-analysis data.

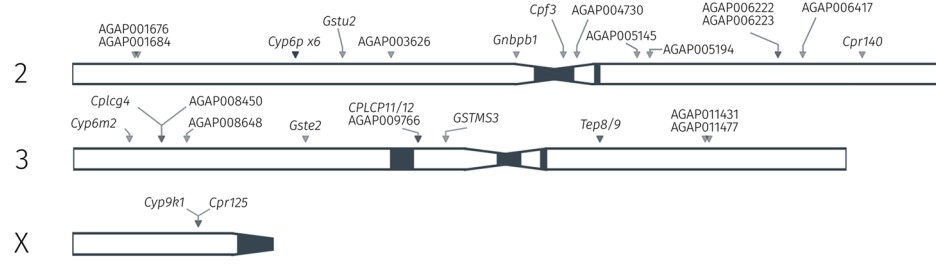

(Supplementary Data 1 and 2). Included in the over-expressed set (cut-off of fold change >2, median counts >5), are 65 and 78 genes respectively. Within these lists, the known pyrethroid metaboliser *CYP6P3*[45] (the ortholog of *CYP6P9a/b* in *An. funestus*) had a median and mean fold change of >6.8, demonstrating the utility of the meta-analytic approach. Included in both lists were other cytochrome P450s including *CYP6P4, CYP9K1, CYP6M2, CYP6Z2, CYP6Z3* and *CYP6P5*, all of which have previously been linked to pyrethroid resistance across both species[21,22]. Interestingly, *CYP4H17* which has never been explored for a role in pyrethroid resistance, appears 7th (mean) and 11th (median) with a median fold change of 3.47. Similarly appearing high across both mean and median rankings are cuticular related proteins *CPLCG4, CPR150* and *CPLCA3*, a venom allergen (AGAP006417), an alkaline phosphatase (AGAP001684), gustatory receptor 49 (AGAP001169) and a protein with no known function (AGAP009327). In total, 50 genes are present in both the median and mean over-expressed sets including one carboxylesterase, eleven cytochrome P450s, three glutathione-S-transferases and one UGT from the major detoxification enzyme families (Fig. 6A). Additionally, nine cuticular proteins, one D7 salivary gland protein and two hexamerins are present from gene families previously linked to resistance[31,46]. Two genes related to odorant or taste detection are also present.

We then performed gene set enrichment analysis on the top 5% of overexpressed genes based on median fold-changes (419 genes) (Supplementary Table 4). We performed enrichment on both GO terms and protein (PFAM) domains; significant enrichment terms are displayed in Fig. 6D, categorised by broad resistance mechanism. In expectation with known gene families involved in insecticide resistance, we observe highly significant enrichment for many GO terms relating to detoxification, such as oxidoreductase activity (fdr-corrected $p$ value = 8e-16), iron ion binding ($p$ = 4.75e-15), heme-binding ($p$ = 6.8e-14), monooxygenase activity ($p$ = 9e-14), glutathione activity ($p$ = 0.001) and glutathione metabolic process ($p$ = 0.001), and the 'P450' ($p$ = 1.6e-16), 'GST_N_3' ($p$ = 0.002) and 'GST_C' ($p$ = 0.004) protein domains. In addition, cuticular-related genes were also enriched, with the GO terms structural constituent of cuticle ($p$ = 1.1e-23), chitin binding ($p$ = 2.3e-09), chitin metabolic process ($p$ = 1.3e-8), fatty acid elongase activity ($p$ = 0.04), and the protein domains 'Chitin_bind_4' ($p$ = 3.2e-20), 'CMB_14' ($p$ = 1.3e-18), and CPCFC ($p$ = 0.03). Finally, a large group of terms were related to sensory perception, including the GO terms 'sensory perception of smell' ($p$ = 1.4e-6), odorant binding ($p$ = 2.4e-6), olfactory receptor activity ($p$ = 0.0002), sensory perception of taste ($p$ = 0.018), taste receptor activity ($p$ = 0.04), and the protein domains '7tm_6' ($p$ = 0.00003) and '7tm_7' ($p$ = 0.03), which indicate odorant and gustatory receptors.

We then explored highly down-regulated genes based on median fold-changes (Fig. 6C). Amongst the most down-regulated genes was *CYP307A1*, a known regulator of ecdysone synthesis[47]. In addition to hormone related transcripts *CLIPA13* and *Galectin 6* are heavily down-regulated. Zero GO terms were enriched based on the 5% lowest median fold-changes. If we instead use the 5% lowest mean fold-changes, we observe significant enrichment for sodium channel activity ($p$ = 0.00013), sodium ion transport ($p$ = 0.00059) and sodium ion transmembrane transport ($p$ = 0.0044) the target sites for pyrethroids.

## Consistently differentially expressed genes

We then explored genes which were significantly up and down-regulated most consistently. After filtering genes with very lowly expressed genes, only two genes were upregulated in 22 out of 28 RNA-Sequencing experiments, *CYP6P3* and *CYP9K1. GSTD3* and *CYP6P4* were upregulated in 21 out 28. Figure 6B shows all 18 genes significantly over-expressed in at least 19 out of 28 experiments, and includes *CYP4H17, CYP4H18, CYP6M2, GSTE2*, a UGT and a trypsin *TRYP4* (Supplementary Table 5). This set of genes were enriched for seven P450-related GO terms, including oxidoreductase activity ($p$ = 1.68e-09) and heme binding ($p$ = 5.66e-09), and the PFAM domain 'p450' ($p$ = 9.19e-10).

In contrast, only eight genes showed consistent down-regulation in at least 19 out of 28 experiments (Supplementary Table 5), including two opsin genes, *GPROP4* and *GPROP6*, and a molecular chaperone HtpG gene (AGAP006961). The genes are enriched in seven GO terms relating to light detection, such as photoreceptor activity (p = 0.029), visual perception (p = 0.029) and phototransduction (p = 0.039).

## Detoxification genes

Twenty-four genes from the P450, GST and carboxylesterase families show either a median or mean fold change of >2 (Supplementary Table 6). Eighteen of those genes were cytochrome P450s, and nine P450s are significantly upregulated in at least 19 out of 28 of all datasets. The cytochrome P450 family most represented is the *CYP6* family, orthologous to the *CYP3* clade in humans, responsible for the vast majority of xenobiotic metabolism. The *CYP6Ps, CYP6M2* and *CYP9K1* which have been repeatedly associated with insecticide resistance are all heavily implicated and are located in regions of the genome under selection (Fig. 5). Additionally, four members of the *CYP4* family appear to be overexpressed across multiple resistant populations, particularly *CYP4H17*, but also *CYP4H16, CYP4H18*, and *CYP4C28*.

Similarly, four out of 27 GSTs (Fig. 6) have high mean or median fold-changes with one, *GSTD3* showing consistently significant overexpression. Of these, *GSTE2* has been shown to be involved in pyrethroid resistance[40], whilst *GSTE1, 2* and *5* have been implicated in DDT resistance[48]. Of the other phase two and three detoxification families, two UGTs (AGAP006222 and AGAP011564) and two carboxylesterases (*COEAE6O* and *COEAE8O*) are expressed with high mean or median expression or consistently over-expressed (Fig. 6).

Pyrethroid metabolising P450s with contrasting patterns of expression within species were also explored. For example, *CYP6P3* is overexpressed in all but two *An. coluzzii* populations; the lab-selected resistant and a field population from Niger. Similarly, *CYP6AA1* and *CYP6M2* are largely overexpressed in *An. coluzzii*, excepting Niger, Nigeria. These data demonstrate that complete convergence of P450 overexpression cannot be expected along a continental scale.

## Cuticular proteins

113 cuticular proteins were included in the full *gamb_colu_arab_fun* analysis; after filtering for lowly-expressed genes, only 61 of these remained. 20 of these genes showed a mean or median fold-change of above two in resistant populations, and although none showed significant over-

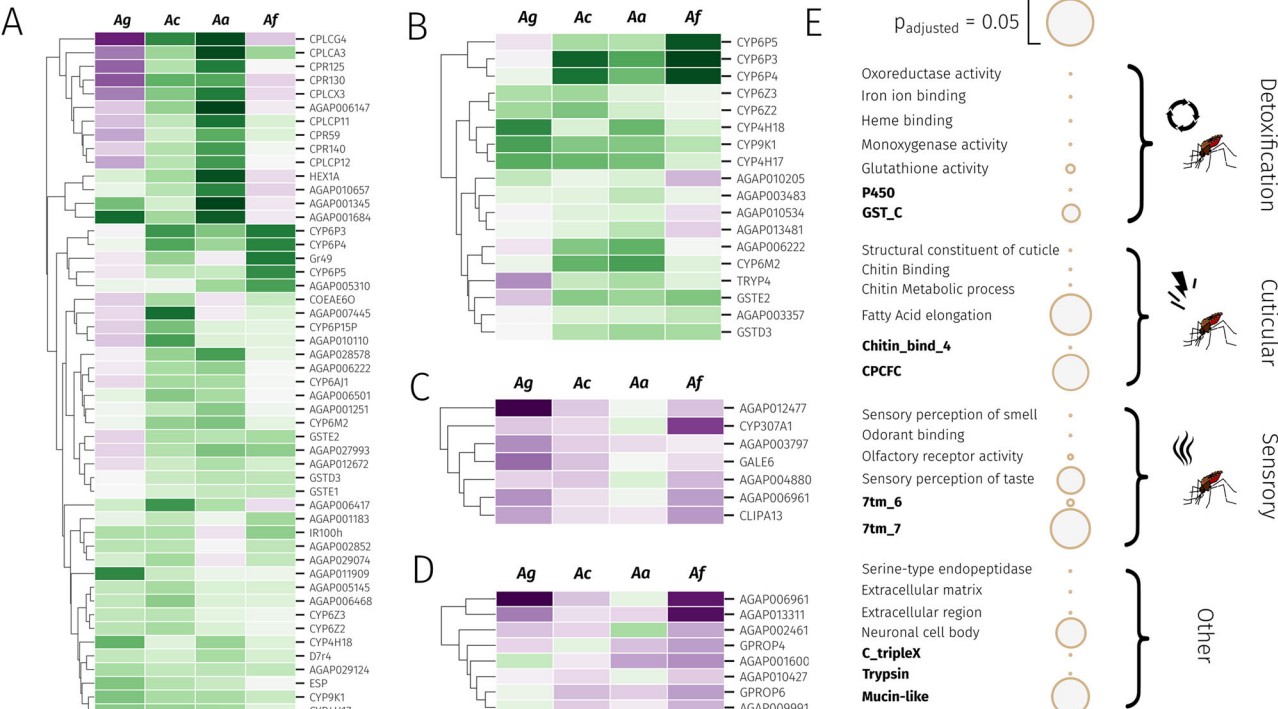

**Fig. 6 | Resistance candidates based on gene expression and gene set enrichment analyses. A** Top genes based on the intersection of genes with mean and median fold-change as above 2, filtered for lowly expressed genes. Ag: *An. gambiae*, Ac: *An. coluzzii*, Aa: *An. arabiensis*, Af: *An. funestus.* **B** Consistently over-expressed genes. Genes that show significant (*p* < 0.05) and positive log2 fold-changes in more than 19 out of 28 experiments. **C** Most down-regulated genes based on median-fold

changes. **D** Most consistently down-regulated genes. Genes that show significant and negative log2 fold-changes in more than 19 out of 28 experiments. **E** Significant GO terms and PFAM domains from gene set enrichment analysis (hypergeometric test) for the top 5% of overexpressed genes based on median fold-changes. Green shows up-regulation of genes whilst purple shows down-regulation, intensity shows relative levels of differential expression.

expression in 19/28 experiments, nine showed upregulation in at least 16, including *CPLCX3, CPR130, CPAP3-C, CPR16, CPR140, CPR59, CPR79, CPLCP12* and *CPR81* (Supplementary Table 7).

### Expression of neuronal-related genes
Gene-set enrichment analysis of GO terms on the 5% most down-regulated genes for median expression revealed significant enrichments in synapse (15 genes), sodium channel activity (10 genes) and chemical synaptic transmission (13 genes). The primary target site for pyrethroid insecticides, the *Vgsc*, is not present in the full *gamb_colu_arab_fun* analysis, but investigation of the secondary analyses shows sporadic down-regulation in *An. coluzzii* and *An. funestus* but not *An. gambiae*, suggesting that changes in expression of this gene are unlikely to be playing a role in resistance. Interestingly, *Ace-1* expression demonstrated up-regulation in almost all *An. coluzzii* and *An. gambiae* populations whilst it is down-regulated in all 5 of the *An. funestus* populations. The gene coding for the beta-2 subunit of the nicotinic acetylcholine receptor is significantly overexpressed in 11 out of 28 experiments.

### Other a priori resistance candidates
A number of genes and gene families have recently been implicated in insecticide resistance, including CSPs, hexamerins, alpha-crystallin, the transcription factor *Maf-S* and d7 salivary proteins[30,31,33,46]. Of the eight chemosensory proteins, just three remained in the full *gamb_colu_arab_fun* analysis, *CSP3, CSP4* and *SAP1*, none of which showed high mean or median fold-changes across four species. Similarly, three hexamerins were included in these data, all three of which showed high mean fold changes ranging from 2.64 to 3.4. Further exploration revealed that overexpression was observed in all *An. arabiensis* and the majority of *An. gambiae* experiments, with the orthologs of *AGAP001345* and *AGAP001657* being overexpressed in two (Uganda and FUMOZ) out of five *An. funestus* datasets. Just two a-crystallins were present in the data, neither of which

showed an expression-association with resistance. The transcription factor *Maf-S* was previously identified as up-regulated across multiple *An. gambiae* s.l populations in the meta-analysis of microarray data[31]. It also shows similarly consistent overexpression in the RNA-Sequencing data, with 12 out of 28 populations exhibiting significant up-regulation.

### Transcriptional regulation of insecticide resistance
Previous work using transcriptomic time-course data revealed putative transcription factors regulating the expression of transcripts post-pyrethroid exposure[49]. Here, we used the GENIE3 algorithm in the Grenadine package to re-capitulate a gene regulatory network utilising the 28 RNA-sequencing and including 31 microarray experiments. After filtering score for the top 5th percentile (Supplementary Data 3), 50,966 interactions were predicted. Firstly, we investigated *Maf-S* due to its prior link with insecticide resistance[33]. 269 putative interactions were identified, including the pyrethroid-resistance related transcripts *CYP6M2, CYP6Z3, CYP9J4*, two ABC transporters and three GSTs including *GSTMS1*. To validate the utility of this approach, microarray data published from a *Maf-S* knockdown[33] was compared with the model predictions; 83 out of the predicted 269 genes were present on the list. We then carried out a permutation analysis, and out of 1000 permutations only 18 had 83 or more overlaps (p = 0.018) giving confidence to the model predictions.

To determine which transcription factors are likely to play a role in resistance, enrichment analysis was performed on the predicted interaction partners for each transcription factor: *Rootletin* was associated with genes significantly enriched for oxoreductase activity (*p* = 0.00943) and was linked with *CYP6P3, CYP6P4* and *CYP6P5, CYP6AA1, CYP6M2* and *CYP6Z4*; genes predicted to interact with *Su(H)* were similarly enriched in oxoreductase activity (*p* = 9.19e-3); *Antp, Cbt* and *Sug* were significantly enriched in cellular response to stress (*p* = 5.19e-4; 3.83e-4; 2.30e-8), *Asciz* and *Org-1* to sensory response to chemical stimulus (*p* = 5.56e-4; 1.69e-4), *E(spl)*

*m3-HLH* and *Grh* with structural constituent of the cuticle ($p = 8.36e-5$; 1.71e-2); *exex* with acetylcholine metabolic process ($p = 3.57e-3$); *Grau, Scrt* and *Toy* with G protein-coupled receptor activity ($p = 1.47e-5$; 1.01e-2; 2.46e-6); HLH3B with neurogenesis ($p = 4.01e-2$); *Onecut* with synapse organisation ($p = 5.26e-3$); *Row* with response to stress ($p = 8.97e-6$) and *TFAM* with electron transport chain (1.25e-7). A number of transcription factors were enriched in higher order functions, such as RNA metabolic processes, gene expression and aromatic compound metabolic processes.

## Discussion

Despite the wide availability of transcriptomic data for major malaria vectors, no study to date has analysed these data across the four primary vectors, *An. gambiae, An. coluzzii, An. arabiensis* and *An. funestus*. In this study we performed a meta-analysis of these data, demonstrating clear convergence of molecular mechanisms of known insecticide resistance genes, as well as discovering novel candidates showing high levels of over-expression across the divergent species. We present a user-friendly python package, AnoExpress, which allows users to explore, visualise, and analyse the transcriptomic meta-analysis data.

Integrating differential expression and single nucleotide polymorphism (SNP) data has the potential to enhance the discovery of novel resistance markers[41,50]. We integrated whole-genome sequence data from the Ag1000G with our transcriptomic meta-analysis, estimating signals of positive selection in wild mosquitoes, and locating the intersection of genes that both are highly overexpressed and are located at sites of selection signals. As over-expression of genes is a common mechanism which can confer insecticide resistance, cis-acting mutations which are likely to increase the expression of nearby genes are expected to be under positive selection. Indeed, cis-acting factors have been identified on a triple-mutant haplotype around *CYP6AA1* in East African *An. gambiae*[23], and in *An. funestus* driving the expression of *CYP6P9a/b*[51].

The transcriptomic meta-analysis recovered many known resistance genes from metabolic gene families, including *CYP6P3* (the *CYP6P9a/b* ortholog), *CYP6P4, CYP6AA1, CYP9K1, CYP6Z2, CYP6Z3, CYP6M2* and *Gste2*[21–23]. Each of these cytochrome P450s have been shown to directly metabolise insecticides[22,52] as well as being associated with selective sweeps in field populations[18,32]. Additionally, several detoxification genes showing consistent overexpression have recently been implicated in gene duplication events in the *An. gambiae* complex, including *CYP9K1, CYP6AA1, COEAE60* and *GSTE2-4*[23,53]. The appearance of these genes in both candidates with consistently high mean/median fold change, as well as those consistently expressed over multiple datasets demonstrates the power of this approach. *CYP4H17* is a striking example of a cytochrome P450 that has been overlooked for further validation and is a promising candidate for functional validation in both *An. gambiae s.l* and *An. funestus*. Examples of differing directionality of expression within species were also seen in within the cytochrome P450s known to actively metabolise pyrethroids, indicating that although the same gene families are often overexpressed across multiple populations, the same patterns shouldn't be applied holistically to all populations across Africa.

In addition to detoxification genes, we see a signature of oxidative stress response. For example, *Maf-S* is involved in the *Nrf2-cnc* pathway, which induces expression of stress response genes upon changes to the oxidative stress levels of the cell[33]. Interestingly, *Maf-S* has been shown to have a role in pyrethroid resistance and was identified from consistent overexpression in microarray datasets[33] and is similarly consistently overexpressed in the RNA-Sequencing data, whilst another member of the pathway, *Keap1*, is upregulated in *An. funestus* from Ghana. In addition to this characterised pathway, protein *DJ-1* (AGAP000705) and *Catalase* are consistently over-expressed, both these genes play a role in protection against oxidative stress[54]. These data are consistent with previous publications indicating that insecticide exposure induces an oxidative stress response[55,56].

Signatures of expression in candidates related to cuticular hydrocarbons were also present, which may indicate a change to lipid processing as recently proposed[57,58] or be related to a thicker cuticle[25]. Genome-wide expression

scans highlighted multiple gene clusters of cuticular protein families as outliers of gene expression, including the *CLPCA, CPLCG, CPLCP* and *CPR* families, which is reflected in the large number of these genes in the top candidate gene lists based on median and mean fold-changes. AGAP010368, an ortholog to a gene involved in fatty acid alpha-oxidation in *Drosophila*, a long chain fatty acid CoA ligase (AGAP009159) and a fatty acid elongase (AGAP003195) are consistently over expressed. The large number of cuticular proteins (20/113) highly expressed and the lack of those consistent across all populations may suggest high levels of redundancy.

We observe little differential expression of the major target site genes, such as the target of pyrethroids the *Vgsc* and dieldrin *Rdl*. As previously reported, the expression profile of the OP and carbamate target *Ace-1* is an exception, often upregulated in *An. coluzzii* and *An. gambiae* yet showing negative fold-changes *An. funestus* experiments. It is worth noting that all five *An. funestus* experiments included in this study use the same susceptible comparator strain, FANG, and so this result could be a peculiarity of this laboratory strain. We find regular over-expression of the gene coding for the beta-2 subunit of nicotinic acetylcholine receptor (AGAP010057), recently associated with Pirimiphos-methyl resistance in a genome-wide association study[12].

As well as physiological changes which directly confer resistance to insecticides, the potential for mosquitoes to avoid contact with insecticides and exploit humans when they are least protected is a serious threat to malaria vector control. Mosquito behaviour is likely to be driven in part by gene expression[59] and transcriptomic studies may allow us to identify candidate genes involved in insecticide-resistant behaviours. This is of particular interest given the difficulty in measuring behavioural resistance in wild mosquito populations. Interestingly, enrichment analyses highlighted that our expression candidate genes were enriched for genes involved in olfactory processes. Gustatory receptor 49 (AGAP001169) shows a median fold-change of 3.39, the highest of any olfactory receptor in our data, and is a promising candidate for behaviourally-linked insecticide resistance. The odorant receptors 13 (AGAP009396) and 9 (AGAP008333) also show high median fold-changes of 2.73 and 2.71, respectively. It is, however, important to note that adaptation of susceptible strains to insectary rearing could cause the downregulation of olfactory genes, and so contribute to these signals.

We also identify candidate genes from gene families not previously associated with resistance in malaria vectors. The ortholog of *neural lazarillo* (AGAP009281) has a high median fold-change, a gene which is known to contribute to longevity, stress resistance and behavioural change through the *JNK* stress response pathway in *Drosophila*[60]. Several venom allergens also exhibit high median fold-changes. Interestingly, *Venom allergen 5* has been associated with resistance in *Culex* through dsRNA and over-expression in cells[61]. These genes are poorly understood but a number have been found in the salivary gland proteome where they have a diverse range of functions, with none currently being characterised[62].

Trans-acting regulation plays an important role in modulating gene expression to insecticide stress in insects[33,63–65]. To explore whether we could identify transcription factors involved in insecticide resistance, we inferred a gene regulatory network. The predicted interactors of several transcription factors were enriched in GO terms related to insecticide resistance. Amongst them is *Rootletin*, which is known to play a role in behavioural response in *Drosophila*[66] and was previously linked to resistance as a key hub in response to insecticide exposure[49], whilst *Grau* has been linked to pyrethroid resistance in *Aedes* populations[67]. Other transcription factors enriched in insecticide resistance-related terms that are consistent with known roles in *Drosophila* includes *Cabut*, which has been shown to regulate stress response[68], *E(spl)m3-HLH* which is linked with stress granules[69], *Grh* that is involved in cuticular repair[70], *Exex* is involved in neuronal function[71], *Onecut* with synapse organisation[72] and *Tfam* is a mitochondrial transcription factor, with a role in oxidative stress response[73].

We also explored the role that gene expression plays in shaping levels of genetic diversity across the genome. Protein evolution is constrained by purifying selection, which acts to prevent deleterious mutations from spreading in a population[74]. Previous studies have shown a relationship

between higher gene expression and a slower accumulation of deleterious mutations[50]. We find a similar relationship in *Anopheles* mosquitoes, with the most highly expressed genes displaying lower rates of pN/pS, as well as a reduction in measures of genetic diversity, and replicate this effect across three mosquito species. *Anopheles gambiae s.l* populations exhibit extremely large effective population sizes, which should accelerate the removal of deleterious mutations. Between species differences support this, as *An. arabiensis* displays lower nucleotide diversity, but higher rates of pN/pS for the most highly expressed genes.

In this study, we present a holistic overview of signatures of gene expression in the major malaria vectors *Anopheles gambiae s.l* and *Anopheles funestus*. As well as establishing relationships between gene expression and whole-genome sequence data, we demonstrate the importance and convergence of detoxification genes, as well as highlighting putative transcription factors and novel gene families involved in insecticide resistance. Despite the power in this study there are several limitations that should be considered; *Anopheles* have a gene/loss factor five times that of *Drosophila*[14] resulting in many gene families with one-to-many orthologs which have been excluded from this study, which results in a loss of data. In addition, the *An. funestus* data is from one experiment, and therefore is likely to have significant batch effects. Next, here we take 'insecticide resistance' as an overall phenotype because the populations grouped in this study have differential resistances to different classes, although all are pyrethroid resistant. Finally, we utilise read count files from published studies and compare within studies only to avoid extraneous confounding factors; however, this will lead to slight variations in expression data due to different methodologies used to extract the counts and precludes inter-study direct comparisons. Even after taking these limitations into count, these data highlight novel functional validation and genomic surveillance targets for malaria vector control in Africa.

## Methods

### RNA-sequencing data

All RNA-sequencing data published comparing resistant and susceptible members of *An. gambiae, An. coluzzii, An. arabiensis* and *An. funestus* were retrieved from Google Scholar in April 2022. Of the 28 resistant populations studied, 15 are *An. coluzzii*, 3 are *An. gambiae*, 5 are *An. arabiensis* and 5 are *An. funestus* (Supplementary Table 1)[34–40]. As multiple studies used the same susceptible populations due to the widespread resistance found in endemic settings, replicates of the same populations across studies were present (Supplementary Table 1). Count files were retrieved from authors of each paper and combined to form one large counts file. VectorBase was used to retrieve orthologs from each species to *An. gambiae* PEST and the counts of one-to-many relationships were averaged.

### Differential expression analysis

The data was then normalised with DESeq2 v1.26[75] using the estimateSizeFactors, estimateDispersions and the varianceStabilizingTransformation methods. Differential expression analysis (DEA) was performed with DESeq2 v1.26 in R v3.6.3. Due to batch effects and large differences in library depth between the included RNA-Sequencing studies, we only performed DEA within each experiment, comparing the susceptible (control) to the resistant strain (case). Positive fold changes indicate over-expression in the resistant strain. Hypothesis testing was performed with the DESeq2 wald test.

### Microarray data

We integrated Microarray data from an earlier meta-analysis[31] into AnoExpress. As this was performed at the transcript level rather than genes, we averaged log2 fold change and adjusted p-value data across transcripts for a given gene to match the AnoExpress meta-analysis.

### Gene-set enrichment analysis

Gene set-enrichment was performed using the hypergeometric test implementation in SciPy, incorporated into AnoExpress. GO terms were extracted from VectorBase and PFAM domains from Grau-Bové et al.[76,77]

### Genomic clustering

Using the *gamb_colu_arab* analysis we calculated and averaged the Pearson's correlation between log2 fold changes for each neighbouring gene pair in the dataset, and then calculated the mean pearsons R across all pairs. We excluded *An. funestus* due to differences in synteny compared to *An. gambiae s.l.* To determine whether this value was higher than expected based on chance, we randomly permuted the positions of all genes in the genome 1000 times, calculated the Pearsons correlation between neighbouring genes, and calculated the mean. We considered the effect significant if the fraction of permutations showing a more extreme value than the actual true value was below 0.05.

### Genetic diversity

Using log2 count data from the *gamb_colu_arab* analysis, we calculated the median counts across all samples, susceptible or resistant, and binned genes into 5% percentiles of expression level. We excluded *An. funestus*, as sufficient whole-genome sequence data is not yet publicly available. We selected three representative cohorts from across the geographic range of each species to analyse from the Anopheles 1000 genome project[32]; *An. gambiae* from Hauts-Bassins; Burkina Faso, Gbadolite; Democratic Republic of Congo, and Nagongera; Uganda, *An. coluzzii* from Hauts-Bassins; Burkina Faso, *Tiassale; Cote D'Ivoire, and Turkana County, Kenya* and *An. arabiensis* from Hauts-Bassins; Burkina Faso, Nagongera; Uganda, and Muleba; Tanzania, collected between 2012 and 2019. We randomly selected 50 individuals from each cohort. Using segregating sites only, we calculated the number of non-synonymous and synoynmous sites for each transcript, using the first transcript annotation for each gene, up to -RC. We calculated the overall CDS length per transcript and calculated the number of synonymous or non-synoynmous mutations per 1000 bp of CDS. We also calculated nucleotide diversity and Wattersons Theta across the entire length of the gene (including introns), using scikit-allel v1.2.1[78]. Populations described in further detail[23,79–81].

### Genome-wide selection scan (GWSS) data

We integrate data from H12 genome-wide selection scans[44] from the selection-atlas. More information on the GWSS and peak-calling algorithm can be found in Supplementary Text 2.

### Genome-wide expression scans

We calculate a sliding window mean of log2 fold-change data, for the *gamb_colu_arab_fun* analysis, with a window size of 10 genes and a step of 2 genes.

### Statistics and reproducibility

All code used for the paper is available on Github (https://github.com/sanjaynagi/AnoExpress) or as part of the Supplementary Text provided with the manuscript.

### Reporting summary

Further information on research design is available in the Nature Portfolio Reporting Summary linked to this article.

## Data availability

Results from the differential expression analyses are stored in the github repository for the AnoExpress python package - https://github.com/sanjaynagi/AnoExpress, and can be explored in the cloud with a series of Google Colaboratory notebooks, or on user's local machines. Source data for the main figures in the manuscript is stored on figshare (https://doi.org/10.6084/m9.figshare.28921100.v2).

All RNA-sequencing data is available from the publications listed in Supplementary Table 1. The authors declare that all other data supporting the findings of this study, are available within the article and its Supplementary Information files.

## Code availability

All code used for this study is available on GitHub: https://github.com/sanjaynagi/AnoExpress or in the provided Supplementary Texts.

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

## Acknowledgements
This study was funded by a Deutsches Zentrum für Infektionsforschung grant (TTU 03.705) to VAI. SCN was funded by an MRC CASE studentship (MR/R015678/1) and a National Institute of Allergy and Infectious Diseases grant (NIAID R01-AI116811 to Martin Donnelly and David Weetman). The authors would like to thank all those who freely gave count data from prior publications, including Louisa Messenger, Dieunel Derilus, Pie Müller, Nadja Wipf, Charles Wondji, Sulaiman Ibrahim, Jack Hearn, Thomas van Leeuwen and Wannes Dermauw. We also thank Hilary Ranson, David Weetman and Alistair Miles for feedback on the manuscript.

## Author contributions
Conceptualisation, analysis, paper drafting (VAI); Conceptualisation, analysis, creation of python package, paper drafting (SCN).

## Funding

## Competing interests
The authors declare no competing interests.
