## [Peer Review file · Communications Biology]

A multi-omic meta-analysis reveals novel mechanisms of insecticide resistance in malaria vectors

Corresponding Author: Dr Victoria Ingham

This manuscript has previously been submitted to another journal. This document only contains information relating to versions considered at Communications Biology.

Version 0:

Reviewer comments:

Reviewer #1

(Remarks to the Author)

This study reports on using different datasets to better understand the underlying molecular mechanisms of insecticide resistance. The analysis focused on the main African malaria vectors. The paper is well written and clear, and only small minor suggestions are made below.

- 1) Line 52: Please confirm if this should be CYP6 insect clade or if this should read the CYP6 family in insects.
- 2) Line 64: Is there any additional referenced that can be included against these species or is these the only studies available? For example, is there only one study on *An. arabiensis*?
- 3) Line 105: Consider rephrasing this to read: "Twenty-eight resistant populations were included in this study from *An. coluzzii* (n=15), *An. gambiae* (n=3) *An. arabiensis* (n=5) and 5 population from *An. funestus*."
- 4) Line 114, etc.: Format numbers as per 34. "8599" should read "8,599." Ditto for the rest.
- 5) Figure 1: Four *Anopheles arabiensis* populations are indicated, but in line 105, five populations were mentioned. The numbers for *An. coluzzii* are also different. Please confirm this.
- 6) Line 137: confirm if this is Figure 1A or if this should be Figure 2A.
- 7) Figure 4, number figure with A and B as per figure legend.
- 8) Line 369: if this is a colony name, it should be in capital letters for FUMOZ.
- 9) References: Reference 1,3: should read World Health Organization. Some references seem incomplete and journal names are missing, ref 6, reference 7 (was published in *Parasite and Vectors* (Sinka, M.E., Bangs, M.J., Manguin, S. et al. The dominant *Anopheles* vectors of human malaria in Africa, Europe and the Middle East: occurrence data, distribution maps and bionomic précis. *Parasites Vectors* 3, 117 (2010). <https://doi.org/10.1186/1756-3305-3-117>), etc. please cross check all references.

Reviewer #2

(Remarks to the Author)

In this work, Nagi and Ingham performed a meta-analysis to characterize gene-expression profiles from published transcriptomic studies across two malaria vector complexes and establish relationships with whole-genome sequence data. They have integrated this data with previous microarray meta-analysis and whole genome data from the *Anopheles* 1000 genome project. Finally, the analyzed data is presented within a Python package called "AnoExpress," where the users can explore gene expression analysis and reproduce some of the analysis performed in this study. As a results, the authors have identified known and novel candidate genes associated with insecticide resistance. The integration of WGS data and transcriptomic data showed a correlation between nucleotide diversity and gene expression, showing an inverse relationship between genetic diversity and gene expression. I think the meta-analysis approach was quite comprehensive and will be of interest to the scientific community. However, I have some major comments and concerns about the integration of the data to the meta-analysis and some results that the authors consider as 'novel' in this study.

Major comments:

1. In L543-L544, you mentioned that the read count files were retrieved from the authors of each paper and combined to form a large count file, which was used as the database for the differential gene expression meta-analysis. Checking the different studies mentioned in Supplementary Table 1, I noticed that the read counts were not generated using the same approach. Some of them counted the number of 'fragments' (read pairs) assigned to protein coding genes, while others counted just the single "reads". I think that this will create a large variability in the database that doesn't have anything to see with the batch effect introduced by different studies, sequencing facilities, or library preparation kits used. For example, if I pick a dataset from any of the papers mentioned in Supplementary Table 1 and computed read count by using the two approaches mentioned above separately (A1= counting single 'reads'; A2=counting 'fragments'), most of the genes in A1, will be 2-fold change overexpressed compared to A2 for the same study. This bias could not be normalized by the custom normalization performed during the RNA-seq analysis. It will be good to convince the readers how the difference in read counting approach of the merged database will not affect the results presented in this study. Yes, you have mentioned that you performed differential gene expression analysis only within each experiment, due to batch effect and large differences in library size between the included studies (L550-L553). However, this sounds like repeating the analysis of these studies, while the purpose of the meta-analysis should be facilitating comparison between studies and finding core resistant makers.

2. L132-L143: Is there any reason to not show in the PCAs the dataset clustered by studies? It would be good to add a second legend in the PCAs plot (even in supplementary table) to show the samples clustered by different study. Since the featureCount table used for this meta-analysis contains read count generated through different approaches, we cannot claim that the clustering pattern observed in the current PCA reflect the true gene expression patterns between and withing groups. Please clarify how the difference in the read count approach will affect the structure of the data used for this analysis.

3. L232-L233: The number of candidate genes reported in the text is not consistent with the number genes found in Supplementary Table 2. Please, Clarify!

4. About correlating genetic diversity (from WGS data) and gene expression data (from transcriptomic data) (L188-L191). While this is an interesting and elegant approach, but I understand that the differences in genetic background of the populations from which the 2 types of datasets were generated will highly affect the correlation observed. By example, the nucleotide diversity calculated from WGS data of *An. arabiensis* from Tanzania was correlated with gene expression data generated from the same species, but from a different population (Burkina Faso and Ethiopia). We expect a large difference in genetic diversity between these two populations that may be driven by geographical distance, climate and other environmental pressures. While I am quite skeptical about the nature of this data for the correlation, the Author need to document how consistent the correlation observed will be compared if the WGS and the RNA-seq data were generated from the same mosquito populations.

5. L578-L580: When you calculated the median read count, it is not clear in the text if you only include resistant groups or the susceptible were added as well. Please clarify.

6. L4334-L436: The authors highlighted that finding overexpression CYP4H17 in the insecticide resistant mosquitoes as novel result. However, this gene was clearly highlighted in the Figure 7 of <https://doi.org/10.3390/insects13030247> (which is included in this meta-analysis) to be highly up regulated in pyrethroid resistant strain of several *An gambiae* s.l populations. Additionally, I did a quick check at this study <https://doi.org/10.1016/j.ibmb.2021.103655> (also part of the meta-analysis), this gene was not tested for differential expression analysis in any of the pairwise comparison; this happened more likely because this gene was filtered out due to low transcription signals (few fragments assigned to genes), which invalidate this gene as candidate maker. Though, this statement and any discussion associated to it should be removed.

7. For the paragraphs in L342-L346, L348-L358, please provide the supplementary table that support all the data mentioned in the text.

8. L282-283: You have performed GO enrichment analysis in only top 5% of overexpressed genes. Is there any specific reason to not compute GO enrichment analysis on all the over expressed genes and highlight the top enriched GO terms instead? In my understanding, to have a better insight of the metabolic processes or metabolic pathways that are activated due to the insecticide exposure, a GO or Kegg pathway enrichment analysis of all the over-expressed genes is more relevant than doing this for only the top ones.

9. The combined read count table used as input for meta-analysis should be provided as supplementary table a way that readers/users could reproduce the analysis and check for feature count details for gene of interest.

10. About AnoExpress platform: Very good work where user can explore gene expression across ~35 experiments used in this study. Where is the video tutorial you provided as you said in L123? One of the best functionalities of this platform is the fact that user can provide a list of gene of interest and see the expression pattern. However, I tried to upload my list of genes but, it was not working. I may not do it correctly; this is why a video tutorial would be very helpfull

Minor comments

1. References for two datasets are not provided in Supplementary Table 1
2. Line 137: Figure 1A should be Figure 2A
3. Line 147: Figure 1B should be in Figure 2B
4. L157-L159: provide references for this statement.

Version 1:

Reviewer comments:

Reviewer #1

(Remarks to the Author)
No additional comments

Reviewer #2

(Remarks to the Author)

The authors have satisfactorily addressed or discussed most of my comments and concerns. I appreciated the efforts of the authors in reviewing the manuscripts, computing new analysis and presented a clearer and a more updated version of this manuscript. I have carefully read the updated manuscript and the rebuttal letter, and I have highlighted the followings for the consideration of both editorial board and Authors:

1. About the response #1:

I completely understand the computational workload problem, but I am not convinced that the 'difference in batch effect and library size' between studies and the difference in approach used for feature count (count 'reads' or 'read pairs') would present a similar issue in downstream analysis. The first is inherent to high throughput RNA sequencing and inter-study comparative analysis; its effect could be mitigated using the data normalization approach and p-value combination techniques as described here. The second issue was described in my previous comment. Additionally, the exercise of counting reads using three different tools and correlating their fold change is fantastic. Still, it does not address my comments/concerns about bias toward counting 'reads' or 'fragments.' This strong correlation observed was expected and suggests that we can easily integrate feature count from different tools for a meta-analysis, but it does not suggest that we can integrate 'read' count and 'fragment' count in such meta-analysis. While, the authors have discussed this approach here, but this new analysis of the authors did not address my comments/concerns.

Finally, I would like to highlight that the idea of computing this RNA-seq meta-analysis is amazing, but I still think the intent of computing inter-study meta-analysis with the normalized (from the same species or from orthologous genes) data would have a higher impact and would allow to get better insight in deciphering gene expression associated with insecticide resistance than repeating the 'within-study' analysis.

2. About the response #2 : from the PCA , the authors mentioned that there was no evidence of data clustering between the two technologies (L145-L149), suggesting the results are comparable. However, this could not be clearly seen by the readers, since in the PCA we have the data points are colored by species. As I mentioned in my previous review, I highly recommend adding the layers for 'sturdy' and 'methodology' (RNA-seq and microarray) to the PCA. The readers will see the different factors that shape the clustering pattern observed.

3. I wonder if the authors found some detoxification genes that were significantly upregulated in some experiments but downregulated in others, for the same species. I don't know if I miss this in the manuscript, but this piece of info would be very informative and justify the importance of such meta-analysis.

Version 2:

Reviewer comments:

Reviewer #2

(Remarks to the Author)
I sincerely apologize for the delay in submitting this review.

This is my third review of the manuscript by Nagi and Ingham, titled "A multi-omic meta-analysis reveals novel mechanisms of insecticide resistance in malaria vectors." The authors have thoughtfully addressed the majority of the comments raised during the first two rounds of review and have provided detailed explanations of their decisions in the rebuttal letter, which is greatly appreciated.

While I do not fully agree with all of the authors' arguments, I am overall satisfied with the substantial improvements made to the manuscript and the quality of the work. I find the approach used to integrate multi-omics datasets to investigate molecular markers of insecticide resistance is novel and valuable. The resources generated through this study will be a useful asset to the scientific community.

Just a minor comment:

It was difficult to read the label axis of the Supplementary Figure 1

We thank the reviewers for their time and input. We have addressed the comments below, including performing additional analyses to allay the concerns of reviewer two over utilising count files from authors and correlations between RNAseq and WGS approaches. All other comments have been addressed in italics below the reviewers' comments.

Reviewers' comments:

Reviewer #1 (Remarks to the Author):

This study reports on using different datasets to better understand the underlying molecular mechanisms of insecticide resistance. The analysis focused on the main African malaria vectors. The paper is well written and clear, and only small minor suggestions are made below.

1) Line 52: Please confirm if this should be CYP6 insect clade or if this should read the CYP6 family in insects.

Changed as suggested, thank you.

2) Line 64: Is there any additional referenced that can be included against these species or is these the only studies available? For example, is there only one study on *An. arabiensis*?

There are other studies building on these – such as RNAseq that then highlights these p450s but these are the original papers showing a direct link with resistance beyond expression or reduced diversity of candidate genes.

3) Line 105:” Consider rephrasing this to read: “Twenty-eight resistant populations were included in this study from *An. coluzzii* (n=15), *An. gambiae* (n=3) *An. arabiensis* (n=5) and 5 population from *An. funestus*.”

We have now rephrased this.

4) Line 114, etc.: Format numbers as per 34. “8599” should read “8,599.” Ditto for the rest.

This has been changed throughout.

5) Figure 1: Four *Anopheles arabiensis* populations are indicated, but in line 105, five populations were mentioned. The numbers for *An. coluzzii* are also different. Please confirm this.

Corrected in the text

6) Line 137: confirm if this is Figure 1A or if this should be Figure 2A.

Changed to 2A, thanks.

7) Figure 4, number figure with A and B as per figure legend.

Corrected

8) Line 369: if this is a colony name, it should be in capital letters for FUMOZ.

Changed.

9) References: Reference 1,3: should read World Health Organization. Some references seem incomplete and journal names are missing, ref 6, reference 7 (was published in Parasite and Vectors (Sinka, M.E., Bangs, M.J., Manguin, S. et al. The dominant Anopheles vectors of human malaria in Africa, Europe and the Middle East: occurrence data, distribution maps and bionomic précis. Parasites Vectors 3, 117 (2010). <https://doi.org/10.1186/1756-3305-3-117>), etc. please cross check all references.

Thank you for spotting this – this is due to migrating from Mendeley to EndNote and has been corrected.

Reviewer #2 (Remarks to the Author):

In this work, Nagi and Ingham performed a meta-analysis to characterize gene-expression profiles from published transcriptomic studies across two malaria vector complexes and establish relationships with whole-genome sequence data. They have integrated this data with previous microarray meta-analysis and whole genome data from the Anopheles 1000 genome project. Finally, the analyzed data is presented within a Python package called “AnoExpress,” where the users can explore gene expression analysis and reproduce some of the analysis performed in this study. As a results, the authors have identified known and novel candidate genes associated with insecticide resistance. The integration of WGS data and transcriptomic data showed a correlation between nucleotide diversity and gene expression, showing an inverse relationship between genetic diversity and gene expression. I think the meta-analysis approach was quite comprehensive and will be of interest to the scientific community. However, I have some major comments and concerns about the integration of the data to the meta-analysis and some results that the authors consider as ‘novel’ in this study.

Major comments:

1. In L543-L544, you mentioned that the read count files were retrieved from the authors of each paper and combined to form a large count file, which was used as the database for the differential gene expression meta-analysis. Checking the different studies mentioned in Supplementary Table 1, I noticed that the read counts were not generated using the same approach. Some of them counted the number of ‘fragments’ (read pairs) assigned to protein coding genes, while others counted just the single “reads”. I think that this will create a large variability in the database that doesn’t have anything to see with the batch effect introduced by different studies, sequencing facilities, or library preparation kits used. For example, if I pick a dataset from any of the papers mentioned in Supplementary Table 1 and computed read

count by using the two approaches mentioned above separately (A1= counting single 'reads'; A2=counting 'fragments'), most of the genes in A1, will be 2-fold change overexpressed compared to A2 for the same study. This bias could not be normalized by the custom normalization performed during the RNA-seq analysis. It will be good to convince the readers how the difference in read counting approach of the merged database will not affect the results presented in this study. Yes, you have mentioned that you performed differential gene expression analysis only within each experiment, due to batch effect and large differences in library size between the included studies (L550-L553). However, this sounds like repeating the analysis of these studies, while the purpose of the meta-analysis should be facilitating comparison between studies and finding core resistant makers.

Thank you for raising this issue – we agree that re-aligning the original files would have eliminated some of these confounders. We chose to use count files provided by authors to minimise computational workloads, with the knowledge that this would mean that only within study comparisons could be performed. Further, the difference in batch effects and library sizes would have presented the same issue with between-study comparisons. As the reviewer points out, we agree that it may affect the read counts themselves; however, as we only do within-study comparisons, the method of counting will not have a large effect on the fold-change or p-values in our results. To address these comments and confirm our approach is valid, we have added an experiment in which we took read data from the Busia study (Nagi et al., Kisumu vs Busia Survivors), and counted reads with Kallisto, HISAT2-featureCounts, and HISAT2-htseqcount. We then performed an identical differential expression analysis on these data with DESeq2, and compared the resulting log2 fold changes. Pearsons correlations on log2 fold change data between the three tools were high (> 0.922). Based on the high-correlations observed here, we are confident that the methods of read counting will not substantially affect our conclusions. We have added these findings in lines 116-121 and included methodology in the supplementary text. The use of DEseq2 on acquired count files ensures the fold changes and p values are as consistent as possible; thus, although this seems like a repeat, it was necessary to perform a meta-analysis on fold changes and p values.

2. L132-L143: Is there any reason to not show in the PCAs the dataset clustered by studies? It would be good to add a second legend in the PCAs plot (even in supplementary table) to show the samples clustered by different study. Since the featureCount table used for this meta-analysis contains read count generated through different approaches, we cannot claim that the clustering pattern observed in the current PCA reflect the true gene expression patterns between and within groups. Please clarify how the difference in the read count approach will affect the structure of the data used for this analysis.

Supplementary Figure 1 shows the PCA requested here, being performed on RNAseq and microarray fold changes, thus eliminating the impacts on counts.

3. L232-L233: The number of candidate genes reported in the text is not consistent with the number genes found in Supplementary Table 2. Please, Clarify!

Thank you for spotting this. We had uploaded the wrong file; we have now updated Supplementary Table 2 to the correct version.

4. About correlating genetic diversity (from WGS data) and gene expression data (from transcriptomic data) (L188-L191). While this is an interesting and elegant approach, but I understand that the differences in genetic background of the populations from which the 2 types of datasets were generated will highly affect the correlation observed. By example, the nucleotide diversity calculated from WGS data of *An. arabiensis* from Tanzania was correlated with gene expression data generated from the same species, but from a different population (Burkina Faso and Ethiopia). We expect a large difference in genetic diversity between these two populations that may be driven by geographical distance, climate and other environmental pressures. While I am quite skeptical about the nature of this data for the correlation, the Author need to document how consistent the correlation observed will be compared if the WGS and the RNA-seq data were generated from the same mosquito populations.

We chose those populations as they were representative populations in the Ag1000g, ie they show patterns which are true of the species generally across the species' range. This approach was recommended by Dr Alistair Miles, lead of the Ag1000g. At the time of writing, there was not publicly available data from the three species from the same location, therefore we were forced to use cohorts from different locations. We selected populations which showed no signs of inbreeding and typical species ancestry. To address your concerns, we have now used three cohorts instead of one in our estimates of genetic diversity, utilising cohorts from across the species range and produced almost identical results (Figure 4)

Broad patterns of gene expression are very similar within a species (as seen even between closely related species in Figure 2), and will not differ enormously between locations, even if these are very far apart within the species range, hence the comparison between rna-seq and wgs from different. If WGS and rna-seq data were taken from the same populations, we would only expect the correlations between diversity and expression to be stronger.

5. L578-L580: When you calculated the median read count, it is not clear in the text if you only include resistant groups or the susceptible were added as well. Please clarify.

This was calculated across all populations, susceptible or resistant. We have clarified.

6. L4334-L436: The authors highlighted that finding overexpression CYP4H17 in the insecticide resistant mosquitoes as novel result. However, this gene was clearly highlighted in the Figure 7 of <https://doi.org/10.3390/insects13030247> (which is included in this meta-analysis) to be highly up regulated in pyrethroid resistant strain of several *An. gambiae* s.l populations. Additionally, I did a quick check at this study <https://doi.org/10.1016/j.ibmb.2021.103655> (also part of the meta-analysis), this gene was not tested for differential expression analysis in any of the pairwise comparison; this happened more likely because this gene was filtered out due to low transcription signals (few fragments assigned to genes), which invalidate this gene as candidate maker. Though, this statement and any discussion associated to it should be removed.

We agree with the reviewer that this candidate has been highlighted in -omics data before; however, it is not a known resistance gene and has never been explored further for functional validation, neither in metabolism assays nor transgenics, or in genomic data when looking for CNVs or reduced diversity etc. This has been clarified. The reviewer's comments about read counts were also checked and this does not appear lowly expressed (majority of populations have read counts in the 100s).

7. For the paragraphs in L342-L346, L348-L358, please provide the supplementary table that support all the data mentioned in the text.

Thanks, we have now added supplementary tables for detoxification and cuticular genes.

8. L282-283: You have performed GO enrichment analysis in only top 5% of overexpressed genes. Is there any specific reason to not compute GO enrichment analysis on all the over expressed genes and highlight the top enriched GO terms instead? In my understanding, to have a better insight of the metabolic processes or metabolic pathways that are activated due to the insecticide exposure, a GO or Kegg pathway enrichment analysis of all the over-expressed genes is more relevant than doing this for only the top ones.

There are two main types of enrichment analysis; here we perform over-representation analysis, which tests (often using the hypergeometric test, as previously published) whether any go terms/kegg pathways are over-represented within a subset of the data (i.e differentially expressed genes). We used 5% top and bottom genes based on median fold change to define these subsets. 5% was a relatively arbitrary threshold which we could use for 'most-overexpressed' and 'most-under expressed' gene lists. The reviewer is absolutely correct that in 'normal' RNAseq experiments, this is done on a larger list but with similar arbitrary cut-offs such as 2x over-expressed and a p value of 0.05 (ie outside 95% of the observed data). We cannot apply that here due to the integrated nature of the approach. The second approach, closer to what the reviewer suggests is a gene set enrichment analysis (gsea) which ranks ALL genes based on a statistic (for example, fold-change, or p-value), and compares the distribution of go terms/kegg pathways in that list to a null distribution. We did not consider this approach here due to the data having relatively conservative fold-changes due to median/mean-based approaches.

9. The combined read count table used as input for meta-analysis should be provided as supplementary table a way that readers/users could reproduce the analysis and check for feature count details for gene of interest.

The read count tables are available here

https://github.com/sanjaynagi/AnoExpress/blob/main/results/log2counts.gamb_colu_arab_fun.tsv ; this is stated in the data availability statement.

The analysis can be reproduced directly from

<https://colab.research.google.com/github/sanjaynagi/AnoExpress/blob/main/workflow/notebooks/differential-expression-meta-analysis.ipynb>

10. About AnoExpress platform: Very good work where user can explore gene expression across ~35 experiments used in this study. Where is the video tutorial you provided as you said in L123? One of the best functionalities of this platform is the fact that user can provide a list of gene of interest and see the expression pattern. However, I tried to upload my list of genes but, it was not working. I may not do it correctly; this is why a video tutorial would be very helpfull

Apologies, we have now added the video tutorial to the documentation home page <https://sanjaynagi.github.io/AnoExpress/landing-page.html>

Minor comments

1. References for two datasets are not provided in Supplementary Table 1

We have added zenodo references to the Supplementary Table 2, as the manuscript for that data has not been published yet.

2. Line 137: Figure 1A should be Figure 2A

This has been corrected

3. Line 147: Figure 1B should be in Figure 2B

This has been corrected.

4. L157-L159: provide references for this statement.

Reference added

Reviewer #2 (Remarks to the Author):

The authors have satisfactorily addressed or discussed most of my comments and concerns. I appreciated the efforts of the authors in reviewing the manuscripts, computing new analysis and presented a clearer and a more updated version of this manuscript. I have carefully read the updated manuscript and the rebuttal letter, and I have highlighted the followings for the consideration of both editorial board and Authors:

1. About the response #1:

I completely understand the computational workload problem, but I am not convinced that the 'difference in batch effect and library size' between studies and the difference in approach used for feature count' (count 'reads' or 'read pairs') would present a similar issue in downstream analysis. The first is inherent to high throughput RNA sequencing and inter-study comparative analysis; its effect could be mitigated using the data normalization approach and p-value combination techniques as described here. The second issue was described in my previous comment. Additionally, the exercise of counting reads using three different tools and correlating their fold change is fantastic. Still, it does not address my comments/concerns about bias toward counting 'reads' or 'fragments.' This strong correlation observed was expected and suggests that we can easily integrate feature count from different tools for a meta-analysis, but it does not suggest that we can integrate 'read' count and 'fragment' count in such meta-analysis. While, the authors have discussed this approach here, but this new analysis of the authors did not address my comments/concerns.

Finally, I would like to highlight that the idea of computing this RNA-seq meta-analysis is amazing, but I still think the intent of computing inter-study meta-analysis with the normalized (from the same species or from orthologous genes) data would have a higher impact and would allow to get better insight in deciphering gene expression associated with insecticide resistance than repeating the 'within-study' analysis.

We have spent time carefully considering the reviewer's comments and agree that in an ideal situation an inter-study meta-analysis would have a higher impact; however, given the differences in the raw fastq files due to a number of extraneous factors, this would also lead to problems. For example, differences in library size, library prep and sequencing batch effects would be impossible to control for. Perhaps more centred on the study here, the computational capacity to do this would be huge and if we were to begin now with full realignment, it would require re-writing of a large proportion of the paper as we would expect results to change if we compare across susceptible comparators, for example. To allay some concerns specific to reads and fragments, we took the Busia vs Kisumu dataset and re-aligned them to the transcriptome. We then used featureCounts to extract both reads and fragments and subsequently did differential expression analysis. We saw minimal difference in generated fold change and the fragments results in ~2x the reads, validating the approach here. Indeed, the correlation for fold changes was 0.9667 and for normalised counts 0.9997. Given this, and the previously mentioned library and sequencing affects, we feel our approach of comparing within studies and then using fold changes is both valid and impactful. We have added this into results on lines 116-

119. *To ensure it is clear that this is a limitation of our study, we have added a few sentences to the discussion in lines 559-562.*

2. About the response #2 : from the PCA , the authors mentioned that there was no evidence of data clustering between the two technologies (L145-L149), suggesting the results are comparable. However, this could not be clearly seen by the readers, since the in the PCA we have the data points are colored by species. As I mentioned in my previous review, I highly recommend adding the layers for 'sturdy' and 'methodology' (RNA-seq and microarray) to the PCA. The readers will see the different factors that shape the clustering pattern observed.

This has now been changed and added as Supplementary files 2A-C.

3. I wonder if the authors found some detoxification genes that were significantly upregulated in some experiments but downregulated in others, for the same species. I don't know if I miss this in the manuscript, but this piece of info would be very informative and justify the importance of such meta-analysis.

Information has been added in the results on lines 356-361 and in the discussion on 460-464.